# Research on Predicting the Turnover of Graduates Using an Enhanced Random Forest Model

**DOI:** 10.3390/bs14070562

**Published:** 2024-07-04

**Authors:** Min Liu, Bo Yang, Yuhang Song

**Affiliations:** 1School of Marxism Studies, Xi’an Polytechnic University, Xi’an 710048, China; lmfengxing@163.com; 2School of Computer Science, Xi’an Polytechnic University, Xi’an 710048, China; 210711024@stu.xpu.edu.cn

**Keywords:** turnover prediction, influencing factors, machine learning, optimized random forest model

## Abstract

The frequent turnover of college graduates is a key factor leading to the frictional unemployment and structural unemployment of youth, which are important research fields concerned with pedagogy, sociology, and management; however, there is little research on the prediction of college graduates’ turnover. Therefore, this study investigated the turnover status of 17,268 college graduates from 52 universities in China, constructed and optimized a random forest model for predicting the turnover of college graduates, and analyzed the influencing mechanism of college graduates’ turnover and the importance of influencing factors. The enhanced random forest model could deal with the unbalanced data and has a higher prediction accuracy as well as stronger generalization ability in predicting the turnover of college graduates. Individual background variables, job characteristic variables, and work environment variables are all important factors influencing whether college graduates resign or not. The top five factors that affect the turnover of college graduates by more than 10% are income level, job satisfaction degree, job opportunities, and job matching degree. The conclusion of this study is conducive to improving the accuracy of turnover prediction, systematically exploring the influencing factors of college graduates’ turnover, and effectively guaranteeing the overall stability of youth employment.

## 1. Introduction

The labor market has long-standing structural contradictions, with labor supply exceeding the demand of enterprises, which can easily lead to unemployment. As the main force of labor, youth often face greater unemployment risks [1]. The ‘Global Employment Trends for Youth 2020: Technology and the future of jobs’ report released by the International Labor Organization [2] shows that the unemployment rate of global youth (between the ages of 15 and 24) is about 13.6%, and that the instability of youth employment has increased rapidly in recent years [3,4]. According to data released by the National Bureau of Statistics of China in October 2022, the unemployment rate of youth aged from 16 to 24 was 17.9% in September. College graduates, as a key group in youth employment, enter the labor market for the first time and generally face the dilemma of frictional unemployment, influenced by multiple factors such as the coronavirus pandemic, the development of new business formats, changes in workers’ concepts, and intensified supply–demand contradiction in the labor market. The frequent turnover of graduates in a short time after graduation is an important manifestation of frictional unemployment. The high turnover rate of college graduates not only aggravates the degree of frictional unemployment and enlarges its duration, but also leads to a sudden rise in the pressure of high-quality employment, keeping employment stable and boosting job creation; therefore, the prediction and evaluation of college graduates’ resignations are not only critical measures with which to effectively alleviate the frictional unemployment of college graduates, but also an objective requirement to promote the higher quality and more stable employment of college graduates.

College graduates’ turnover is comprehensively affected by various factors, including internal and external factors as well as subjective and objective factors. Currently, there is still no consensus on the critical influencing factors. Therefore, predicting the turnover of college graduates is challenging. In recent years, with the rapid development of artificial intelligence research, machine learning algorithms have been widely used in both different scientific fields and daily life. In particular, the random decision forest algorithm has been popular in many behavioral analysis fields that are difficult to predict and explain because of its excellent predictive performance. To this end, the random forest algorithm in the machine learning field can be combined with the prediction of college graduates’ turnover to enhance risk monitoring and early warning as well as improve the corresponding handling procedure of college graduates’ resignations. Based on the survey data of 17,268 college graduates from 52 colleges and universities and their turnover status half a year after graduation in Shaanxi Province, an enhanced random forest model to predict the turnover of college graduates is established after processing imbalanced data. In this way, the influencing factors and influencing trends in college graduates’ resignations are deeply mined. Thus, to actively respond to the potential factors affecting college graduates’ turnover under the background of stable employment and to prevent as well as resolve the large-scale frictional unemployment risk of college graduates, this research provides a theoretical method and supporting technology that realize the transition from data to knowledge and also help for forecasting and making decisions. This research aims to provide theoretical and practical references for ensuring the overall stability of the employment situation of college graduates.

## 2. Literature Review

As an essential method of individual mobility, resignation usually presents as a turnover behavior or process. Because of the sudden occurrence of resignation behavior and the concealment of the resignation process, scholars at home and abroad focus on the influencing factors and prediction algorithms of resignation.

### 2.1. Influencing Factors of Turnover

The factors that influence turnover intention are complicated, and scholars overseas emphasize exploring the influencing factors of individual turnover by constructing models, such as Mobley’s model of intermediate linkages [5], the Steers and Medway model [6], Sheridan and Abelson’s cusp catastrophe model [7], Lee and Mitchell’s unfolding model [8], the Price–Mueller model [9], and so on. By constructing an influencing factor model of turnover, we can find that the influencing factors of individual resignation mainly include the following three aspects:(1)Individual background variable: Scholars pointed out that turnover intention is the most powerful predictor among predictive variables of actual individual turnover behavior and the direct antecedent variable that best predicts turnover behavior [10,11]. Weisberg and Kirschenbaum found that gender influences employee turnover behavior [12]. They explained the differences in resignation behavior between men and women. When studying the influencing factors of the voluntary resignation of college students in micro, small, and medium enterprises, researchers found that the voluntary turnover proportion of college graduates in literature, history, philosophy, and art was significantly higher than that of science and engineering students. In contrast, the voluntary turnover proportion of college graduates among different groups of family locations and college types was not significantly different [13].(2)Job characteristic variable: Martin showed that relative individuals’ salary is an important factor affecting their turnover [14]. Harald also revealed that there is a negative correlation between employee compensation, as well as benefits, and resignation behavior [15]. Saks and Ashforth found that the mismatch between personal majors and occupations has a great negative impact on turnover and job switching [16]. Roznowski and Hulin found that job satisfaction is an important antecedent variable of turnover behavior and is one of the most effective factors for researchers and managers to predict turnover intention [17]. Kang and Gatling and Kim pointed out that teachers’ turnover negatively correlates with their job satisfaction [18]. If teachers have high job satisfaction, they are more inclined to stay in their work organization rather than leave, which thereby reduces the occurrence of teacher resignation. When analyzing the impact of enterprise organization features on employee turnover, Bluedorn indicated that the more advanced the management level and mechanism system of an enterprise, the more attractive it is to employees and the lower the possibility of employee turnover [19]. Moreover, some scholars believe that places with high levels of economic development and large economic aggregates have more opportunities for occupational mobility compared with places with low levels of economic development and small economic aggregates [20].(3)Work environment variable: Sousa-Poza and Henneberger found that the resignation of employees is significantly related to their working environment [21]. Pfeffer pointed out that an efficient working environment and good working atmosphere can reduce employee turnover behavior [22]. Price believed that job opportunity is an important factor influencing the turnover intentions of employees [9]. When organizations provide employees with more development opportunities to meet individual needs, the job stability and durability of employees will be higher, and they are less likely to leave an organization. Karavardar pointed out that the resignation behavior of employees is greatly negatively related to the development opportunities of employee professional capability and the speed of job promotions [23].

### 2.2. Prediction Algorithm for Turnover

The algorithms for predicting employee turnover in academic research mainly include general statistical algorithms and machine learning algorithms. General statistical algorithms consist of factor analysis, discriminant analysis, and so on. For instance, Ghosh et al. established a discriminant function to determine the best predictive factors for employee resignation by employing discriminant analysis, based on utilizing factor analysis to obtain the influencing factors of resignation [24]. The predictive model based on the general statistical algorithm requires that sample data satisfy characteristics such as a linear relationship or normal distribution, which will greatly reduce the applicability and accuracy of the prediction model to a certain extent.

In recent years, with the striking upsurge of big data, the research on employee turnover prediction has been gradually carried out by means of machine learning methods in academia. Machine learning methods can obtain a rule by data analysis and then make the prediction of the data-belonging fields based on the rule, which can provide a reference for planning and making decisions. For example, based on the idea of the QSIM algorithm, Xia et al. introduced qualitative simulation technology into an employee turnover prediction model by combining the change features of employee turnover behaviors [25]. They used VB programming to simulate and realize virtual experimental research on employee turnover prediction. Punnoose used the HRIS data of a global retailer to compare the extreme gradient boosting (XGB) algorithm with other machine learning algorithms and emphasized exploring the application of the XGB algorithm in employee turnover prediction [26]. The research shows that the XGB algorithm has great advantages in significantly improving prediction accuracy, relatively shortening the running time, and efficient memory usage. Ribes et al. developed a model to predict employee turnover by using the standard machine learning technique [27]. This model not only analyzed the importance of factors affecting employee turnover but also found that an effective way to prevent employee turnover is to allow talent flow of employees between different positions. Zhao et al. focused on analyzing and evaluating the capability of 10 supervised machine learning methods to predict employee turnover, including a decision tree (DT), random forest (RF), logistic regression (LR), gradient boosting decision tree (GBDT), XGB, artificial neural network (ANN), and support vector machine (SVM) [28]. Their results show that different machine learning algorithms can be employed for different sizes and complex human resource datasets to enhance the interpretability of employee turnover models. Karande and Shyamala laid special stress on establishing an integrated employee turnover prediction model that can support machine learning methods such as an SVM, LR, and RF [29]. This model can more accurately predict employee turnover based on weighting individual classification and calculating the weighted average.

In conclusion, machine learning algorithms can significantly improve the efficiency and accuracy of employee turnover prediction compared with general statistical algorithms [30]; however, each machine learning algorithm has certain limitations: the processing of unbalanced data is imperfect, the prediction accuracy needs to be improved, and the ranking of feature importance is missing. Therefore, based on the above-mentioned existing research, individual background variables, job characteristic variables, and work environment variables are selected as the predictors of college graduate turnover to compare the predictive performance of different machine learning methods, LR, RF, SVM, a convolutional neural network (CNN), and Bayesian optimization random forest (BORF), on the modeling effect of college graduate turnover in this study. The importance of the influence of individual background variables, job characteristic variables, and work environment variables on the turnover of college graduates is evaluated, which is expected to provide a basis for the prediction, analysis, and response of the college graduate turnover.

## 3. Research Methods and Design

### 3.1. Research Samples

The data used in this study were from the “Universities Graduates Employment and Entrepreneurship Quality Tracking Survey”, which was based on the online questionnaire survey conducted by the Western China Higher Education Evaluation Center of Xi’an Jiaotong University for 2019 graduates in Shaanxi Province, China. The survey period was from January to April 2020. The respondents are college graduates who were not employed before obtaining their degree. Among the 17,268 college graduates participating in the survey, 13,944 did not leave their jobs within six months, accounting for 80.8%, while 3324 college graduates had the experience of leaving their jobs, accounting for 19.2%. In addition, among the 3289 resigned graduates who continued to answer the “way of resignation”, 3062 graduates resigned voluntarily, accounting for 93.1%, while the proportion of dismissed graduates was 2.5%, and the rate of both of ways of resignation was 4.4%.

### 3.2. Predictive Variables

Predictive variables were measured when the respondents were already employed. The descriptive statistical results of the specific predictive variables are shown in Table 1 and Table 2.

Individual background variables mainly involve gender, type of college, type of major, family location, turnover intention, etc. Gender is divided into male and female; the types of colleges are mainly divided into key universities, general universities, and private undergraduate universities and independent colleges; type of major is classified into two types: science and engineering, and humanities and social sciences; family location mainly includes large- and medium-level cities, towns/county-level cities, and rural areas; and turnover intention refers to the possibility of college graduates’ resignation within six months after graduation.

Job characteristic variables mainly involve income level, job matching degree, job satisfaction, nature of working unit, work area, and so on. Income level refers to all of the discounted cash income of college graduates who participated in the survey, including average monthly salary, bonus, performance commission, and cash welfare subsidy. Job matching degree is the match condition between the work position signed by college graduates and their majors as well as their individual career goals. Job satisfaction refers to the overall satisfaction of college graduates with job position, salary and welfare, geographical location, career stability, and career development in the employment unit. The nature of working unit is generally divided into five categories: party and government organizations, state-owned enterprises, public institutions, foreign-funded enterprises, and private enterprises. The work area mainly includes three regions: eastern regions, central regions, and western regions.

Work environment variables mainly contain job opportunities, working atmosphere, work pressure, and so on. Job opportunities represent the self-development opportunities, training opportunities, job promotion opportunities, and other opportunities provided for college graduates by the work unit. The work atmosphere refers to the atmosphere or environment that is gradually formed in a working unit and has certain characteristics and can be perceived and recognized by college graduates. The specific questions are “I feel the team members are willing to listen to and understand my opinions and suggestions”, “I feel my personal work is important to the team”, and “my team is able to work together efficiently”. Work pressure refers to the stress of college graduates caused by excessive work intensity.

### 3.3. Research Method

In this study, an enhanced random forest model, named Bayesian optimization random forest (BORF), is proposed to predict college graduates’ resignation, which usually includes three procedures: data preprocessing, model construction, and model parameters’ determination.

#### 3.3.1. Data Preprocessing

In the data from the practical survey, the ratio of the number of university graduates who have left their jobs to the number of those who have not left their jobs is roughly 1:4. The data show prominent class imbalance characteristics [31], that is, the number of graduates who left their positions is substantially higher than the number of resigned graduates. Suppose the optimization objective is to maximize the overall classification accuracy; in this case, the model will be biased towards predicting the data of most non-resignation graduates, while the data from the minority resigned graduates are ignored. Therefore, the CTGAN method is adopted to automatically generate the simulated departure data by learning the distribution from the original data so that the dataset achieves class balance [32].

#### 3.3.2. Model Construction

The RF utilizes a voting strategy in ensemble learning to integrate multiple decision trees into a unified prediction model, which can effectively solve classification and regression problems [33]. The random forest can train each decision tree in parallel, so it has a faster training speed [34]. Because the decision tree belongs to a simple prediction model, the RF method has better robustness to data noise, and can effectively prevent overfitting. A schematic diagram of the random forest classification process in this study is shown in Figure 1. First, set the decision tree number, N, included in the random forest. Then, subdatasets will be generated by sampling with replacements from the overall sample and repeated N times. Each subdataset corresponds to a decision tree. Next, take the Gini index as a standard for subtree splitting until the selected feature has been used in the tree or has reached the preset deepest depth [35]. Then, this feature is identified as a leaf node feature, and this decision tree is constructed completely. Finally, the voting strategy is used to integrate the prediction outputs of N decision trees and to form the final classification result of the random forest model.

#### 3.3.3. Model Parameter Determination

The model parameters are usually determined by adopting the grid search strategy; however, when there are many parameters the search speed will be slow and excessive computing resources will be wasted. A Bayesian optimization algorithm can automatically optimize the parameter selection process. Compared with the traditional grid search method [36], it can effectively shorten the training time and improve the performance of the model. The main parameters in the BORF model proposed in this study are all obtained by a Bayesian optimization algorithm, such as the number of decision trees, the factor number in the decision tree, the deepest depth of the tree, the minimum number of samples in nodes, etc.

## 4. Results

### 4.1. Evaluation Criterion of the Model

An effective and reasonable evaluation criterion can accurately evaluate the model performance and guide the selection of the model structure for real data. In this study, to accurately evaluate the classification effect of the random forest model, five evaluation indicators are utilized, i.e., accuracy, precision, recall, $F_{1}$ value, and area under the curve (AUC) value. In addition, *TP* represents the number of graduates who resigned both actually and in the model prediction. *FP* means the number of graduates who did not actually resigned but the model predicted have resigned. *FN* represents the number of graduates who actually resigned but the model predicted did not leave. *TN* is the number of graduates who did not actually leave and the model predicted did not leave. 

The accuracy refers to the proportion of correctly predicted samples to the total number of samples:(1)$$Accuracy=\frac{TP+TN}{TP+TN+FP+FN}$$

The precision of turnover is the ratio of actual resignations to the total samples of resignations in the prediction:(2)$$Precision=\frac{TP}{TP+FP}$$

The recall of resignations is the proportion of the samples that are correctly predicted as resignations to the actual resignation samples:(3)$$Recall=\frac{TP}{TP+FN}$$

The $F_{1}$ value is the harmonic mean of precision and recall:(4)$$F_{1}=2\cdot \frac{Precision\cdot Recall}{Precision+Recall}$$

The AUC value is the area under the receiver operating characteristic (ROC) curve, which can comprehensively reflect the classification performance of the machine learning model. The AUC value is between 0.1 and 1. The larger the AUC value, the better the classification effect of the model.

### 4.2. Prediction Performance of the Model

In the test stage, 85% of the sample data are selected as the training set and the rest are used as the test set. The test environment is a computer with an AMD Ryzen 5 4600H CPU, a main frequency of 3 GHz, and 16 GB of memory. The prediction results of the LR model, RF model, SVM model, CNN model, and the proposed BORF model on the test set are compared and analyzed in Table 3 in this study.

We can observe the following from Table 3: The first is about the time consumption of the models. All the models were run 100 times, and the average running time was calculated. Among the models, the LR model uses the shortest time, with only 0.12 s, and the BORF model ranks second by 0.12 s, while the SVM model and CNN model take a long time to process. The second is based on evaluation indicators of overall accuracy and AUC value. Compared with the other four models, the BORF model has a better performance in classification prediction, and its evaluation indicators are all at a relatively leading level. Thirdly, the prediction effect of the sample data classified as resigned graduates is observed according to the three evaluation indicators: precision, recall, and $F_{1}$ value. The BORF model still performs well in the classification prediction effect of class-imbalanced sample data, while the prediction capability of the RF model and the LR model in the sample data of a small number of leaving graduates is slightly inferior. The fourth is that the BORF model has an overall classification accuracy of 78.6% for the resignation status of college graduates. Among the sample of actual resigned college graduates, the proportion of the correctly predicted resignation is close to 70%. This means that the BORF model has a good application performance in the prediction of college graduates’ resignation, which contributes to the realization of the monitoring and early warning of college graduates’ resignations.

### 4.3. Influencing Factors’ Importance Analysis of Models

During the process of model construction, the data that do not participate in the training of the decision tree are called out-of-bag data. Firstly, the generalization $err1_{i}$, i=1,2,⋯,N of the *i*th decision tress is computed by out-of-bag data. Then, noises are added to characteristic factor x and the values are recorded to calculate the generalization error $err2_{i}$. Finally, the importance of factor x can be obtained by Equation (5). This method is employed to calculate and rank the importance of factors influencing college graduates’ turnover in this study, and the results are shown in Table 4.
(5)$$I_{x}=\sum_{i=1}^{N}(err2_{i}-err1_{i})/N$$

It can be observed from Table 4 that the importance of income level, job satisfaction degree, job matching degree, nature of working unit, and work area from the job characteristic variables accumulates to 50.2%. The significance of individual background variables, involving turnover intention, type of colleges, family location, gender, major, and other factors, totals 31.0%. The influence of the importance of job opportunity, work atmosphere, and work pressure from the work environment variables adds up to 18.8%. Moreover, the relative importance of income level, turnover intention, job satisfaction, job opportunity, and job matching degree in the turnover of college graduates all reach over 10%. Nevertheless, work area, gender, work pressure, and major factors have little influence on the turnover of college graduates, whose relative importance is less than 5%.

## 5. Discussion

### 5.1. Prediction of the Turnover of College Graduates by Bayesian Optimization Random Forest

Compared with logistic regression (LR), random forest (RF), support vector machine (SVM), and the convolutional neural network (CNN), the Bayesian optimization random forest (BORF) is relatively advanced in predicting the factors affecting the turnover of college graduates in terms of accuracy, precision, recall, $F_{1}$, AUC, and other evaluation indexes. The model has strong generalization ability and anti-noise performance, has a better prediction for the class imbalance sample data of college graduates’ turnover, and has an overall classification accuracy of 78.6%. This result fully indicates that BORF is an important tool for monitoring and warning the influencing factors of college graduates’ resignations.

### 5.2. The Effect of Individual Background and College Graduates’ Turnover

It is found that job characteristic variables have a significant effect on the resignation of college graduates in this study. Among these variables, income level plays the most important role in influencing graduate turnover, with an importance of over 15%. Job satisfaction, involving the overall satisfaction of college graduates with their job position, salary and welfare, interpersonal relationships, career stability, and career development, is the second significant factor affecting their turnover behavior, with an important proportion of 13%. These results fully show that the critical factors influencing whether university graduates resign or not are income level and job satisfaction. Moreover, the job matching degree is a quite important influencing factor of graduate resignation, and the importance degree is over 10%. This result suggests that college graduates take the match condition between their work position and major, as well as their individual career goals, into account as the main indicator to weigh whether they resign or not. Differences in the nature of working units also directly influence graduate turnover behavior, with an influencing factor of 7.5%. This result reflects the fact that university graduates value the nature of work units and regard it as a considerable part of a resignation decision.

### 5.3. The Effect of Individual Background on College Graduates’ Turnover

We find that individual background variables are major factors influencing college graduate turnover in this study. Among the individual background variables, many studies usually argue that turnover intention is the most important factor influencing turnover, and consider turnover intention as a mediating variable when analyzing the relationship between job satisfaction, job matching degree, career opportunity, and turnover. This study found that income level has the most significant impact on turnover, and the importance of turnover intention is slightly lower than that of income level. The type of university influences graduate turnover to some degree, of which the importance reaches 5.6%. The result shows that the key universities attach their reputation value to the personal value of university graduates, which prompts graduates to achieve higher quality employment and effectively reduces the possibility of key university graduates’ turnover. Furthermore, the location of college graduates’ family is one of the factors that influence their turnover, and its importance exceeds 5%. This shows that the difference in family social resources is one of the important factors affecting college graduates’ turnover.

### 5.4. The Effect of Work Environment on College Graduates’ Turnover

The working environment is an influencing factor of college graduates’ turnover, which is concluded in this study. Among the working environment factors, job opportunities have a more significant impact on graduate turnover, with an impact of over 10%. This reflects the fact that college graduates think highly of the development opportunities, training opportunities, promotion opportunities, etc., provided by their work units. If the job opportunities provided by college graduates’ work units cannot meet their expected goals, the possibility of graduate turnover will increase significantly.

## 6. Conclusions and Implications

### 6.1. Conclusions

Based on the above research results on college graduates’ turnover prediction and the important analysis of the influencing factors, the following conclusions are mainly drawn in this study:

Compared with the LR model, RF model, SVM model, and CNN model, the proposed BORF model has a higher prediction accuracy and stronger generalization ability for the problem of whether college graduates leave their jobs. The BORF model can effectively process class-imbalanced data, and its prediction effect on the actual resign sample of college graduates is better. The results fully show that the BORF model has a good application performance on the prediction of college graduates’ resignation, which can help improve the early warning and prediction ability of college graduates’ turnover risks.

Using the optimized random forest model to calculate the importance of factors affecting the turnover, this study found that individual background variables, job characteristic variables, and work environment variables are all important factors influencing whether college graduates resign or not, but their impact differs.

Job characteristic variables have the most significant impact on turnover, with a cumulative importance of 50.2%. Among them, income level, job satisfaction, and job matching all have more than 10% importance for turnover. Individual background variables also significantly affect turnover, with a cumulative importance exceeding 30%. Among them, the importance of turnover intention in turnover is more than 13%. Working environment variables have a certain significant impact on turnover, and the cumulative importance is close to 20%. Among them, the importance of job opportunities on turnover is more than 10%.

### 6.2. Implications

According to the above research conclusions, to effectively reduce the turnover rate of college graduates, to alleviate the frictional unemployment dilemma of college graduates, and to make every effort to ensure the overall stability of college graduates’ employment situation, the following implications for policy and practice are put forward in this study:

Establish a systematic, three-dimensional, and dynamic prediction as well as early warning mechanism for college graduates’ resignation. Enhance the capacity to predict and warn about large-scale resignation risks of college graduates. Strengthen the prevention measure of large-scale college graduate resignation to effectively prevent the potential risks caused by large-scale resignation. Realize theoretical methods and supporting technologies that convert data into knowledge and then into prediction and decision making.

Take advantage of income level, job matching degree, and job opportunities to alleviate turnover. The turnover possibility of college graduates can be reduced by establishing a well-developed salary and welfare system, improving the matching degree of professions and jobs, and providing reasonable development opportunities, training opportunities, and promotion opportunities.

Pay attention to the inhibitory effect of job satisfaction on turnover and enhance the job well-being of college graduates. Job position, geographical location, salary and benefits, career stability, career development, and other aspects should be focused on to improve the job satisfaction of college graduates and so to effectively alleviate the turnover intention of college graduates.

### 6.3. Limitations

This study is limited to at least three aspects: Firstly, the findings are based on employed graduate participants in Shaanxi Province of China. Participants from other locations might have produced different results. Secondly, this study only discusses the influence of individual background, job characteristics, and working environment on the turnover of college graduates, without considering the influence of realistic environments and so on. This requires further analysis of the influencing factors of graduates’ turnover on a more multidimensional basis. Thirdly, this study does not consider turnover intention as a mediating variable when analyzing the relationship between salary, job satisfaction, job matching degree, career opportunity, and turnover behavior. In future studies, we will focus on the position of turnover intention among predictive variables and discuss the implications for the accuracy of the optimized random forest model.

## Figures and Tables

**Figure 1 behavsci-14-00562-f001:**
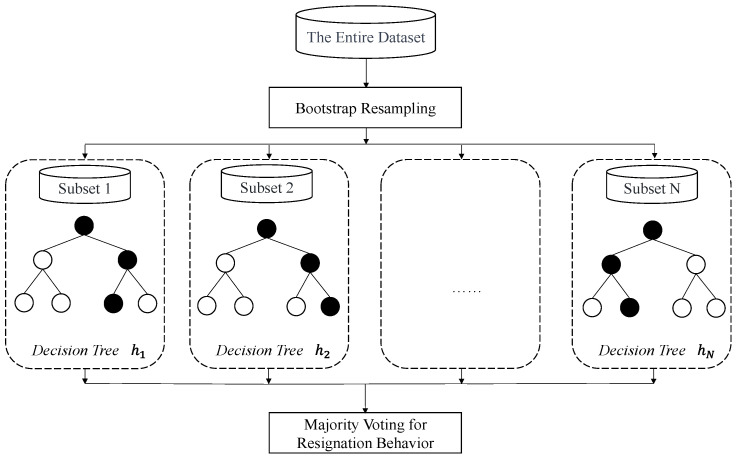
The schematic diagram of the random forest classification process.

**Table 1 behavsci-14-00562-t001:** The descriptive statistical results of the classified predictive variables.

Group	Proportion (%)
Gender	
Male	51.6
Female	48.4
Type of College	
Key university	16.4
General university	59.6
Private undergraduate and independent college	24.0
Major	
Science and engineering	49.5
Humanities and social sciences	50.5
Family Location	
Large and medium-level city	28.3
Towns/county-level city	38.2
Rural area	33.5
Income Level (CNY/Monthly)	
1000–3000	16.8
3001–4000	21.5
4001–5000	23.8
5001–7000	22.3
Over 7000	15.6
Nature of Working Unit	
Party and government organization	4.9
Public institution	19.6
State-owned enterprise	28.1
Foreign-funded enterprise	7.0
Private enterprise	40.3
Work area	
Eastern region	30.6
Central region	7.2
Western region	62.3

**Table 2 behavsci-14-00562-t002:** The descriptive statistical results of scored predictive variables.

Group	Average	Standard Deviation
Job matching degree (scored from 1 to 5)	3.37	0.95
Job satisfaction degree (scored from 1 to 5)	3.50	0.70
Job opportunity (scored from 1 to 5)	3.39	0.86
Work atmosphere (scored from 1 to 5)	3.95	0.68
Work pressure (scored from 1 to 5)	2.28	0.55
Turnover intention (scored from 1 to 5)	2.90	0.90

**Table 3 behavsci-14-00562-t003:** Comparative analysis of the prediction effect of different models on college graduates’ turnover.

Model	Predictive Class	Overall Accuracy (%)	Accuracy (%)	Recall (%)	$F_{1}$ Value	AUC Value	Running Time (s)
LR	Non-resigned	69.1	87.3	73.0	0.79	0.64	0.07
	Resigned		32.7	55.9	0.42		
RF	Non-resigned	73.5	84.6	84.1	0.85	0.59	0.58
	Resigned		34.3	35.4	0.35		
SVM	Non-resigned	67.6	86.9	70.3	0.78	0.63	2.54
	Resigned		31.6	56.3	0.41		
CNN	Non-resigned	69.6	86.1	74.2	0.80	0.62	5.82
	Resigned		32.5	51.0	0.40		
BORF	Non-resigned	78.6	85.1	89.2	0.87	0.69	0.12
	Resigned		35.2	68.1	0.46		

**Table 4 behavsci-14-00562-t004:** Importance analysis of the factors influencing college graduates’ turnover.

Influence Factor	Ranking	Relative Importance	Accumulative Importance (%)
Income level	1	0.1579	15.8
Turnover intention	2	0.1349	29.3
Job satisfaction degree	3	0.1303	42.3
Job opportunity	4	0.1051	52.8
Job matching degree	5	0.1018	63.0
Nature of working unit	6	0.0752	70.5
Type of colleges	7	0.0561	76.1
Family location	8	0.0555	81.7
Work atmosphere	9	0.0502	86.7
Work area	10	0.0371	90.4
Gender	11	0.0331	93.7
Work pressure	12	0.0322	96.9
Major	13	0.0308	100.0

## Data Availability

The data used in this study were from the “Universities Graduates Employment and Entrepreneurship Quality Tracking Survey”, which was based on an online questionnaire survey conducted by the Western China Higher Education Evaluation Center of Xi’an Jiaotong University for 2019 graduates in Shaanxi Province, China.

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
