# Peer review of "Research on Predicting the Turnover of Graduates Using an Enhanced Random Forest Model"

_behavsci, 2024, doi:10.3390/bs14070562_

Round 1

Reviewer 1 Report

Comments and Suggestions for Authors

Scientific Review of the Article: "Research on Predicting the Turnover of Graduates Using an Enhanced Random Forest Model"

Introduction

The article "Research on Predicting the Turnover of Graduates Using an Enhanced Random Forest Model" addresses the prediction of college graduates' turnover using machine learning algorithms, particularly an enhanced random forest model. The authors aim to improve the prediction accuracy and generalization ability in understanding the turnover of college graduates, an essential factor in tackling frictional and structural unemployment among youth.

The study involved a survey of 17,268 college graduates from 52 universities in China. The authors constructed and optimized a random forest model to predict the turnover of these graduates. They identified individual background variables, job characteristics, and work environment variables as critical factors influencing whether college graduates resign from their jobs. The model demonstrated high prediction accuracy and generalization ability, indicating its potential usefulness in early warning systems for graduate turnover. Although the article is generally well structured and clear, there are some critical issues that need to be resolved:

  1. Lack of Employment History Details:
    • Issue: The article does not reference the employment history of the respondents, such as the number of graduates who were already employed before obtaining their degree. This omission leaves a gap in understanding the baseline employment status of the participants.
    • Importance: Knowing how many graduates were employed before graduation could significantly impact turnover predictions. For example, individuals who secured employment shortly after an extensive job search might be less likely to leave their jobs compared to those who found jobs immediately.
  2. Timing of the Survey:
    • Issue: The timing of the questionnaire administration is not specified. There is no indication of how long after graduation the survey was conducted, which is crucial for contextualizing the responses.
    • Importance: The period between graduation and the survey could affect the graduates' job stability and their responses. If the survey is conducted too early, the data might not accurately reflect long-term employment trends and turnover intentions.
  3. Temporal Relationship between Variables and Resignation:
    • Issue: The study lacks clarity on the temporal relationship between predictive variables and resignation events. Specifically, it is unclear when the predictive variables were measured relative to when the employment status (still employed vs. resigned) was recorded.
    • Importance: For predictive variables to be effective, they must be measured when all subjects are still employed. Additionally, variables like turnover intention can fluctuate over time, affecting their predictive power.
  4. Position of Turnover Intention among Predictive Variables:
    • Issue: Turnover intention is treated as a variable on the same level as other predictive factors, despite its nature as a mediating variable influenced by factors such as salary, job satisfaction, career opportunities, …
    • Importance: Turnover intention should be positioned between other predictive variables and the outcome (resignation). High correlations with other variables indicate causal relationships. The authors need to justify statistically the placement of turnover intention alongside other variables and discuss its implications for the model's accuracy.
  5. Definition of "Work Atmosphere":
    • Issue: The article does not provide a clear definition of "work atmosphere" or specify the questions used to assess it.
    • Importance: A precise definition and understanding of the measurement of work atmosphere are necessary to interpret its impact on turnover accurately. Without this, the variable's role in the model remains ambiguous.

Author Response

Manuscript ID: behavsci-3058883

Title: Research on Predicting the Turnover of Graduates Using an Enhanced Random Forest Model

Authors: Min Liu, Bo Yang*, Yu hang Song

Journal: Behavioral Sciences

We are very grateful to Reviewer 1 for the critical and professional review of our manuscript. Reviewer 1 gives us many helpful and constructive comments from the aspect of empirical research, which improve this paper impact and clarity. Please find a point-by-point response to your comments below.

Comment 1: Lack of Employment History Details:

  • Issue: The article does not reference the employment history of the respondents, such as the number of graduates who were already employed before obtaining their degree. This omission leaves a gap in understanding the baseline employment status of the participants.
  • Importance: Knowing how many graduates were employed before graduation could significantly impact turnover predictions. For example, individuals who secured employment shortly after an extensive job search might be less likely to leave their jobs compared to those who found jobs immediately.

Response 1: Thank you for pointing this out. Therefore, we have supplemented the background information of the respondents. The respondents are college graduates who weren’t employed before obtaining their degree. They have been studying in colleges and have no working experience before they obtain their degree.

Comment 2: Timing of the Survey:

  • Issue: The timing of the questionnaire administration is not specified. There is no indication of how long after graduation the survey was conducted, which is crucial for contextualizing the responses.
  • Importance: The period between graduation and the survey could affect the graduates' job stability and their responses. If the survey is conducted too early, the data might not accurately reflect long-term employment trends and turnover intentions.

Response 2: We are grateful for this constructive comment. This survey was conducted of graduates of the class of 2019, who have worked for half a year, and the survey period was from January to April 2020.

Comment 3: Temporal Relationship between Variables and Resignation:

  • Issue: The study lacks clarity on the temporal relationship between predictive variables and resignation events. Specifically, it is unclear when the predictive variables were measured relative to when the employment status (still employed vs. resigned) was recorded.
  • Importance: For predictive variables to be effective, they must be measured when all subjects are still employed. Additionally, variables like turnover intention can fluctuate over time, affecting their predictive power.

Response 3: We agree with this helpful comment, as we have not described clearly the temporal relationship between predictive variables and resignation events in the initial submission. Indeed, during the survey period, 17,268 university graduates were employed. Therefore, all predictive variables were measured six months after obtaining their degree and when respondents were already employed.

Comment 4: Position of Turnover Intention among Predictive Variables:

  • Issue: Turnover intention is treated as a variable on the same level as other predictive factors, despite its nature as a mediating variable influenced by factors such as salary, job satisfaction, career opportunities, …
  • Importance: Turnover intention should be positioned between other predictive variables and the outcome (resignation). High correlations with other variables indicate causal relationships. The authors need to justify statistically the placement of turnover intention alongside other variables and discuss its implications for the model's accuracy.

Response 4: Many thanks for the professional comments. Actually, many studies usually considered the turnover intention as a mediating variable when analyzing the relationship between salary, job satisfaction, job matching degree, career opportunity and turnover behavior. Through the optimized random forest model to calculate the importance of factors affecting the turnover, this study found that: income level has the most significant impact on the turnover, and the importance of turnover intention on the turnover is slightly lower than income level. In future studies, we will focus on the position of turnover intention among predictive variables and discuss the implications for the accuracy of the optimized random forest model.

Comment 5: Definition of "Work Atmosphere"

  • Issue: The article does not provide a clear definition of "work atmosphere" or specify the questions used to assess it.
  • Importance: A precise definition and understanding of the measurement of work atmosphere are necessary to interpret its impact on turnover accurately. Without this, the variable's role in the model remains ambiguous.

Response 5: We are grateful for requesting the definition of "Work Atmosphere". Actually, work atmosphere refers to the atmosphere or environment that is gradually formed in a working unit and has certain characteristics and can be perceived and recognized by college graduates. The specific questions are that “I feel the team members are willing to listen to and understand my opinions and suggestions” and I “feel my personal work is important to the team, my team is able to work together efficiently”.

Reviewer 2 Report

Comments and Suggestions for Authors

Most of the references are obsolete; kindly update them. The conclusion session needs to be improved upon by tailoring the results. Providing implications for policy and practice will add value to this work. 

Comments on the Quality of English Language

 There is a need for the service of a language editor to improve the quality.

Author Response

Manuscript ID: behavsci-3058883

Title: Research on Predicting the Turnover of Graduates Using an Enhanced Random Forest Model

Authors: Min Liu, Bo Yang*, Yu hang Song

Journal: Behavioral Sciences

We would like to thank Reviewer 2 for the careful and thorough assessment of our manuscript. and for offering helpful and constructive comments. We appreciate that addressing your comments has yielded a considerably clearer contribution. Please find a point-by-point response to your comments below.

Comment 1: Most of the references are obsolete; kindly update them.

Response 1: We are grateful for requesting to update the reference. In the revised manuscript, we have added some recent references marked in red.

Comment 2: The conclusion session needs to be improved upon by tailoring the results.

Response 2: We are grateful for this constructive comment. In response, we improve some conclusions based on the analysis results. In this revision, these sentences now read: “Using the optimized random forest model to calculate the importance of factors affecting the turnover, this study found that individual background variables, job characteristics variables, and work environment variables are all important factors influencing whether college graduates resign, but their impact differs. Job characteristic variables have the most significant impact on the turnover, with a cumulative importance of 50.2%. Among them, income level, job satisfaction and job matching are all more than 10% importance for the turnover. Individual background variables also significantly affect turnover, with a cumulative importance exceeding 30%. Among them, the importance of turnover intention on turnover is more than 13%. Working environment variables have a certain significant impact on turnover, and the cumulative importance is close to 20%. Among them, the importance of job opportunities on the turnover is more than 10%.”

Comment 3: Providing implications for policy and practice will add value to this work.

Response 3: Many thanks for the professional comments. In response, we provide the implications for policy and practice. For example, establishing a systematic and dynamic prediction and early warning mechanism for resignation, taking advantage of income level, job matching degree and job opportunities to alleviate the turnover, paying attention to the inhibitory effect of job satisfaction on turnover and enhancing the job well-being of college graduates.

Comment 4: There is a need for the service of a language editor to improve the quality.

Response 4: Thank you so much for professional comments. The manuscript has been revised and re-polished. We hope it will meet the standard of Behavioral Sciences.
